# Evolution of the Surgical Management of Lung Cancer Invading the Spine: A Single Center Experience

**Gabrielle Drevet** [1,*] **, Théo Broussolle** [2] **, Yanis Belaroussi** [1] **, Lucie Duponchelle** [1] **, Jean Michel Maury** [1] **, Renaud Grima** [1] **, Gualter Vaz** [3] **, Clément Silvestre** [4] **and François Tronc** [1]

1    Department of Thoracic Surgery, Lung and Heart-Lung Transplantation, Louis Pradel Hospital, 69677 Lyon, France; yanissbelaroussi@gmail.com (Y.B.); lucie.duponchelle@chu-lyon.fr (L.D.); jean-michel.maury@chu-lyon.fr (J.M.M.); renaud.grima@chu-lyon.fr (R.G.); francois.tronc@chu-lyon.fr (F.T.)
2    Department of Neurosurgery and Spine Surgery, Pierre Wertheimer Hospital, 69677 Lyon, France; theo.broussolle@chu-lyon.fr
3    Department of Surgery, Léon Bérard Cancer Centre, 69008 Lyon, France; gualter.vaz@lyon.unicancer.fr
4    Department of Orthopedic Surgery, Clinique Médico-Chirurgicale des Massues, 69005 Lyon, France; silvestre@chirurgie-rachis-lyon.fr
\*    Correspondence: gabrielle.drevet@chu-lyon.fr

**Abstract:** For patients with locally advanced non-small cell lung cancer invading the spine, induction chemoradiotherapy combined with radical en bloc resection is the key to obtaining long-term survival. With time, our operative technique evolved to a two-step surgery as we experienced numerous perioperative complications during one step surgery. The aim of our study was to assess postoperative morbimortality and long-term survival of both techniques. We retrospectively reviewed all patients who underwent en bloc resection for lung cancer invading the spine between October 2012 and June 2020. Every patient underwent induction therapy. Sixteen patients were included: nine patients were operated on with one step surgery, seven patients were operated on with two step interventions. Twenty-five percent of patients had major perioperative complications and 56.2% of patients had major post-operative complications. Patients in the "one step" group tended to have more perioperative complications whereas patients in the "two step" group tended to have more post-operative complications. Overall 3-year survival was 40% in the one-step and 86% in the two-step surgery group. Although our practice has been improved by two-step interventions, post-operative morbidity remains significant. As long term survivals are encouraging, this type of treatment should still be proposed for highly selected patients, in specialized centers.

**Keywords:** lung cancer; vertebral involvement; locally advanced lung cancer; vertebrectomy; lung cancer invading the spine

## 1. Introduction

Surgical resection is the treatment of choice for early stage non-small cell lung cancer [1]. For patients with locally advanced non-small cell lung cancer invading the spine, multimodal treatment has demonstrated considerable improvements in terms of the long-term survival and control of symptoms [2] as this group of patients often suffer from significant pain and neurologic sequelae. In T4-N0/N1 patients, the treatment should be aggressive in order to maximize the chances of long-term disease control and the most critical point is complete resection with sufficient negative margins. In carefully selected patients, the use of induction chemoradiotherapy combined with radical en bloc resection [3] allows complete resection in a vast majority of patients and offers promising long-term outcomes. Thus, technical aspects of surgical resection have evolved over the past few years. Initially described with a 3-incision approach: a midline posterior vertebral approach, a transclavicular anterior cervical approach for tumors of the thoracic inlet and a posterolateral thoracotomy, this surgery has been progressively simplified [4]. Nowadays, a 2-incision

surgical approach is preferred with a transmanubrial cervicothoracic anterior incision or an extended posterolateral thoracotomy combined with a midline posterior spinal approach. At the beginning of our experience, lung and vertebral resection were realized in the same operating time. However, concerns arose after numerous perioperative complications during one step surgery, particularly hemodynamic instability when positioning the patient from anterior or lateral position to prone position, which can lead to cardiac arrest. So, we switched to a two step surgery by approximately two weeks, i.e., vertebral posterior osteotomy and spine stabilization first and secondly anterior or posterolateral approach to complete vertebral resection with en bloc lung and parietal resection.

The aim of our work was to study the supposed benefit of the modified surgical management of lung cancer invading the spine and to assess the postoperative morbidity, mortality and long-term survival of both approaches.

## 2. Materials and Methods

### 2.1. Study Design

We retrospectively reviewed all patients who underwent en bloc resection of non-small cell lung cancer invading the spine between October 2012 and June 2020 at the Louis Pradel University Hospital (Lyon, France). Every patient underwent a complete pre-operative pulmonary and oncologic evaluation including, at least, computed tomography (CT) scan of the chest, positron emission tomography scan, CT scan or magnetic resonance imaging (MRI) of the brain and pulmonary function tests. Vertebral invasion was assessed by MRI to evaluate the extent of bone infiltration and the extent of foraminal and epidural involvement. Electroneuromyography (ENMG) and invasive exploration of mediastinal adenopathy were not performed routinely but only in selected cases. Electroneuromyography was performed if there was any doubt about peripheral nerve involvement and mediastinoscopy or EBUS was performed in the case of suspicious adenopathy on PET scan. Diagnosis was systematically obtained by percutaneous fine-needle aspiration. For each patient, a multidisciplinary tumor board validated surgical management. As recommended, every patient underwent induction treatment consisting of four cycles of a cisplatin-based doublet chemotherapy with concurrent radiation. The staging was reviewed prior to surgery to confirm operability.

Inclusion criteria were histologic diagnosis of non-small cell lung cancer, no clinical N2 disease, no distant metastases, radiological criteria for spine invasion and neo adjuvant therapy. Exclusion criteria were diagnoses other than primary lung cancer, unresectable tumor, performance status higher than two and disease progression after induction treatment.

The following data were extracted: demographics, medical history, smoking history, symptoms at diagnosis, neo-adjuvant and adjuvant therapy, date and type of surgery, peri- and post-operative complications using Clavien classification, post-operative lung cancer staging, recurrence and final outcome. Data were collected until January 2021.

Tumors were classified and staged according to the 8th edition TNM classification for lung cancer. Response to neo adjuvant treatment was defined as none, partial and complete. Overall survival was estimated from the date of surgery to the occurrence of death. Disease-free survival was estimated from the date of surgery to the date of recurrence diagnosis.

### 2.2. Operative Technique

The surgical technique evolved throughout the study period. Between 2013 and 2016, surgical lung and vertebral resections were performed during the same operating time. Between 2016 and 2020, the surgical approach consisted of a two-step surgery. First, a spinal approach was performed about 15 days (range from 8 to 22 days) before lung resection. A posterior approach of the spine and a specific thoracic incision were systematically performed. Depending on the specific tumor location, a posterior Paulson approach with the patient in lateral decubitus position or a transmanubrial cervicothoracic anterior approach with the patient in supine position for tumors of the cervico-thoracic

junction (C7/T3) were performed (Figure 1). When a posterior Paulson approach was performed, the vertebral resection was initiated and finalized by way of the thoracic incision. When a transmanubrial cervicothoracic anterior approach was undertaken, the vertebral resection was performed in the first place by a posterior approach to the spine through a midline skin incision, in the same operating time or 15 days before lung resection. The posterior approach consists of posterior stabilization and separation of the resection piece, with an osteotomy passing through the vertebral body or the pedicles, depending on preoperative planning through a CT scan. Care must be taken to separate the posterior wall of the vertebra from the dural sheath to minimize the risk of dura mater injury during the en bloc resection of the operative piece. By the anterior approach, the pleural space is entered through the third intercostal space and the first rib is cut anteriorly. Subclavian vessels were freed from the tumor and vertebral osteotomy was completed by the way of the anterior approach. Anterior spine stabilization was also performed if required. In both techniques (anterior or posterior), hilar structures dissection and lobectomy are performed in a standard fashion. Radical lymph node dissection was undertaken. Lung and vertebral resection with involved chest wall were performed en bloc with wide margins. Reconstruction of the posterior and anterior columns of the spine is necessary because of the extensive bone resection, resulting in significant instability. The anterior column is reconstructed with cages (usually made of titanium), placed after the vertebrectomy, and the posterior reconstruction is performed with pedicle screws connected with rods (also made of titanium, or cobalt chrome), placed during the posterior approach to the spine. The arthrodesis obtained is therefore circumferential. Since the use of local bone autograft is contraindicated, bone substitutes or iliac crest bone can be utilized for bone grafting. The levels to be fused posteriorly should be at least two to three levels above and below the corporectomy, taking into account the sagittal balance of the spine. Indeed, it is important to avoid stopping at the cervicothoracic or thoracolumbar junctions, or at the apex of the dorsal kyphosis, in order to limit mechanical complications (in particular pseudarthrodesis, rod breakage or proximal junctional kyphosis).

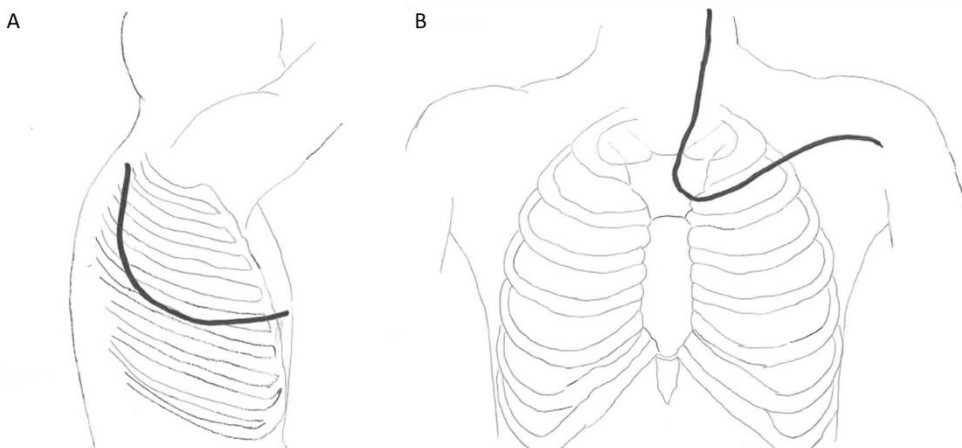

**Figure 1.** Possible thoracic incisions depending on tumor location, represented by a black line. (**A**). Posterolateral thoracotomy extended to the seventh cervical vertebra (Shaw-Paulson incision). (**B**). The transmanubrial cervico-thoracotomy with first rib section (Cormier–Dartevelle–Grunenwald incision).

*2.3. Statistical Analysis*

The population was described globally, then both "one step surgery" and "two steps surgery" groups were described separately. Data were summarized with frequency and percentage for qualitative variables and median [interquartile range] for continuous variables. A survival analysis was performed using Kaplan–Meier methods. Overall survival and

progression-free survival were analyzed and date of surgery was considered as baseline. Analyses were performed using R software version 4.0.3.

## 3. Results

This section may be divided by subheadings. It should provide a concise and precise description of the experimental results and their interpretation, as well as the experimental conclusions that can be drawn.

### 3.1. Patient Population

During the study period, 16 patients (3 females and 13 males) were diagnosed with non-small cell lung cancer invading the spine and were referred to our Thoracic Surgery Department (Table 1). Median age was 56 (range, 44 to 69 years). Half the patients presented with an adenocarcinoma, 37% of patients with squamous cell carcinoma and 13% of patients with poorly differentiated carcinoma.

**Table 1.** Patients characteristics.

|  | *n* = **16 (%)** |
|---|---|
| Gender | |
| male | 13 (81%) |
| female | 3 (19%) |
| Medical history | |
| smoking | 16 (100%) |
| stopped smoking | 11 (58%) |
| Anatomopathology | |
| Adenocarcinoma | 8 (50%) |
| Squamous cell carcinoma | 6 (37%) |
| Poorly differentiated carcinoma | 2 (13%) |
| Neo adjuvant radiotherapy | |
| Chemotherapy | |
| up to 3 cycles | 5 (31%) |
| 3 to 6 cycles | 11 (69%) |
| Radiotherapy | |
| 45 grays | 12 (75%) |
| 66 grays | 3 (19%) |
| Resection | |
| One-step | 9 (56%) |
| Two-step | 7 (44%) |
| pN status | |
| N0 | 14 (87%) |
| N1 | 0 |
| N2 | 2 (13%) |
| Completeness of resection | |
| R0 | 16 (100%) |
| R1 | 0 |
| R2 | 0 |

Every patient received between 2 to 6 cycles of platinum-based induction chemotherapy. All patients except one underwent neo-adjuvant concurrent radiotherapy for a total dose of 45 Gy (ranged from 40 to 66 Gy). After induction therapy, 14 patients had a partial radiologic response and 2 patients had a stable disease.

*3.2. Outcomes*

Overall, 9 patients underwent a one-step procedure and 7 patients underwent a two-step procedure. Five patients operated on with the one-step surgery underwent the posterior Paulson approach and four patients underwent the transmanubrial anterior approach. Every patient operated on with the two-step surgery underwent the transmanubrial anterior approach. The 16 patients of our study underwent upper lobectomy. No pneumonectomy or sub lobar resection was performed. The average number of ribs resected was 3.4 (range, 2 to 5) and the average number of vertebrae resected was 2.6 (range, 1 to 3) (Table 2). Mean estimated blood loss was 1107 mL (range from 300 to 2000) in the one step surgery group and 558mL (range from 250 to 800) for the two-step surgery group. Per operative complications occurred in 4 patients who underwent thoracic and spine approaches in the same operating time only. Three patients had significant blood loss during surgery requiring important blood transfusions. Among them, one patient suffered from uncontrolled venous hemorrhage originating from epidural veins, after posterior osteotomy and release of the spinal canal. He benefited from a rudimentary closure with sponges left inside for quick positioning in a supine position to control hemodynamic instability. He was reoperated on two days later for hemostasis without difficulties. One patient suffered from a cardiac arrest when changing from a lateral to a prone position probably favored by insufficient blood volume replacement. He recovered without aftermath with immediate resuscitation maneuvers. In the two-step surgical management, no post first-step-intervention complications led to postponing the second surgery and no perioperative complications occurred during the second step such as adhesions or hematomas consecutive to the first step.

All patients were considered N0 at the end of the radiologic staging but only 14 patients were N0 on final pathology and 2 patients were incidentally diagnosed with N2 disease. These two patients had only one station N2 involvement. All cases had pathological T4 disease. All patients had complete resection (R0). Post-operative pathologic response to induction treatment was complete, near complete and partial in 7 patients (44%), 4 patients (25%), 5 patients (31%), respectively.

**Table 2.** Surgical characteristics and complications.

|  | One-Step *n* = 9 (%) | Two-Step *n* = 7 (%) |
|---|---|---|
| Number of rib(s) resected | | |
| 2 | 2 (22.2%) | 1 (14.3%) |
| 3 | 1 (11.1%) | 4 (57.1%) |
| 4 | 4 (44.5%) | 2 (28.6%) |
| 5 | 2 (22.2%) | 0 |
| Number of vertebra(e) resected | | |
| 1 | 1 (11.1%) | 0 |
| 2 | 1 (11.1%) | 3 (42.9%) |
| 3 | 7 (77.8%) | 4 (57.1%) |
| Per operative complications | | |
| Hemorrhage | 3 (33.3%) | 0 |
| Cardiac arrest | 1 (11.1%) | 0 |
| Post-operative complications | | |
| Prolonged drainage | 7 (77.8%) | 4 (57.1%) |
| Pneumonia | 9 (100%) | 5 (71.4%) |
| Atrial fibrillation | 2 (22.2%) | 3 (42.9%) |
| Renal insufficiency | 0 | 2 (28.6%) |
| Upper limb paresis | 4 (44.5%) | 3 (42.9%) |
| Cerebrospinal fluid leak | 2 (22.2%) | 0 |
| Neuropathic pain | 3 (33.3%) | 4 (57.1%) |
| In hospital death | 0 | 1 (14.3%) |

In total, every patient had at least a minor post-operative complication. Minor post-operative complications (grade I and II of the Clavien classification of surgical complications [5]) such as prolonged thoracic drainage and pneumonia or atrial fibrillation evolving favorably after adequate medication occurred in 11 (69%), 14 (87%) and 5 (31%) patients, respectively. Major complications (grade IIIa, IIIb, IV and V of the Clavien classification [5]) occurred in 9 patients (56.2%) (Table 3). Major complications included atelectasis requiring flexible bronchoscopy (4 patients 25%), respiratory insufficiency requiring tracheotomy (1 patient, 6.2%), surgical revision due to post-operative empyema (2 patients, 12.5%), surgical revision due to wound infection (3 patients, 18.7%), surgical management of osteosynthesis material infection (1 patient, 6.2%) and surgical revision due to cerebrospinal fluid leak (1 patient, 6.2%). No residual neurologic impairments were observed. One post-operative death (6.25%) occurred in a patient who underwent two-step surgical resection. The patient developed pyocyanic pneumonia progressing to ARDS and septic shock. Respiratory insufficiency with multiple extubation failures, the appearance of renal insufficiency and poorly tolerated atrial fibrillation led to death 56 days after surgery. Compared to two-step surgery, patients operated on with one-step demonstrated a trend towards a higher perioperative complication rate. Inversely, patients operated on with the two-step intervention demonstrated a trend towards more frequent post-operative complications (25% vs. 71.4%). Thus, major complications seemed to occur similarly in both groups. No particular relationship between the number of resected ribs or vertebrae and the type or severity of complications were observed. The mean hospital length of stay for patients who underwent the one-step procedure was 12.8 days vs. 6.1 days for patients who underwent the two-step procedure.

**Table 3.** Post-operative complications according to Clavien classification [5].

|  | One-Step *n* = 9 | Two-Step *n* = 7 |
|---|---|---|
| **Major post-operative complications** |  |  |
| Atelectasis requiring flexible bronchoscopy | 1 (11.1%) | 3 (42.9%) |
| Respiratory insufficiency requiring tracheotomy | 1 (11.1%) | 0 |
| Surgical revision due to post-operative empyema | 1 (11.1%) | 1 (14.3%) |
| Surgical revision due to wound infection | 2 (22.2%) | 1 (14.3%) |
| Osteosynthesis material infection | 0 | 1 (14.3%) |
| Cerebrospinal fluid leak | 1 (11.1%) | 0 |
| **Clavien classification** |  |  |
| Grade I | 1 (11.1%) | 1 (14.3%) |
| Grade II | 4 (44.5%) | 1 (14.3%) |
| Grade IIIa | 1 (11.1%) | 2 (28.6%) |
| Grade IIIb | 2 (22.2%) | 2 (28.6%) |
| Grade IVa | 1 (11.1%) | 0 |
| Grade IVb | 0 | 0 |
| Grade V | 0 | 1 (14.3%) |

In the post-operative period, no patient received adjuvant therapy. The median follow-up period was 25 months (range from 12 to 39 months). Recurrences were seen in 5 (31%) patients and were local in 2 and systemic in 3 patients. A trend towards better survival and less recurrence in the 2-step surgery group was observed. Overall 3-year survival in the whole population of the study was 66% (Figure 2A). The overall 3-year survival in the one-step surgery group was 40% and the overall 3-year survival in the two-step surgery group was 86% (Figure 2B). Both pN2 patients had a recurrence. One patient died within the year after surgery and the other died at 2 years. Disease free survival rate in the whole population of the study was 55% at 3 years (Figure 3A). The 3-year disease free survival rate was 29% in the one-step surgery group and the 3-year disease free survival rate was 86% in the two-step surgery group (Figure 3B).

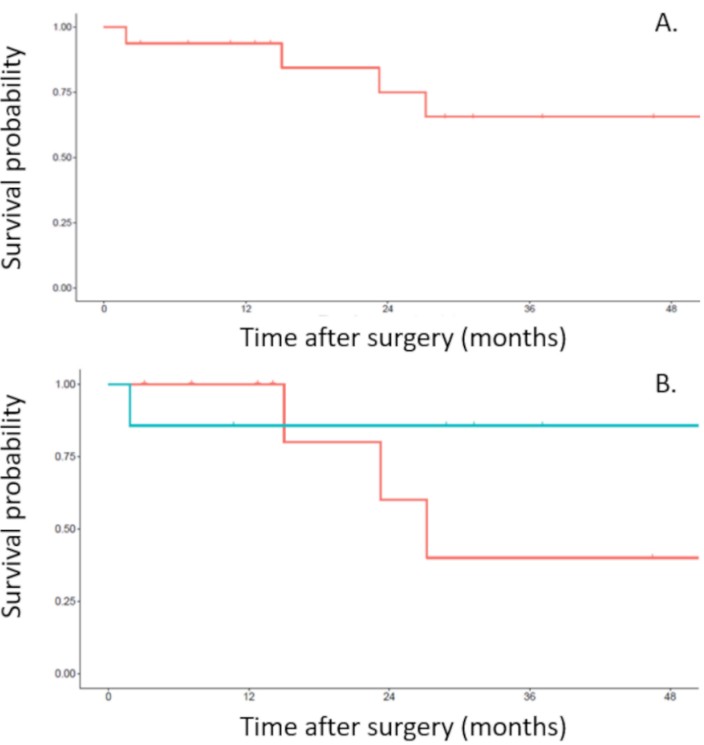

**Figure 2.** Overall survival from the date of en bloc resection of the upper lobe, chest wall and vertebra. (**A**). Overall survival of the whole population of the study. (**B**). Overall survival related to the one-step (red line) or two-step (blue line) approach.

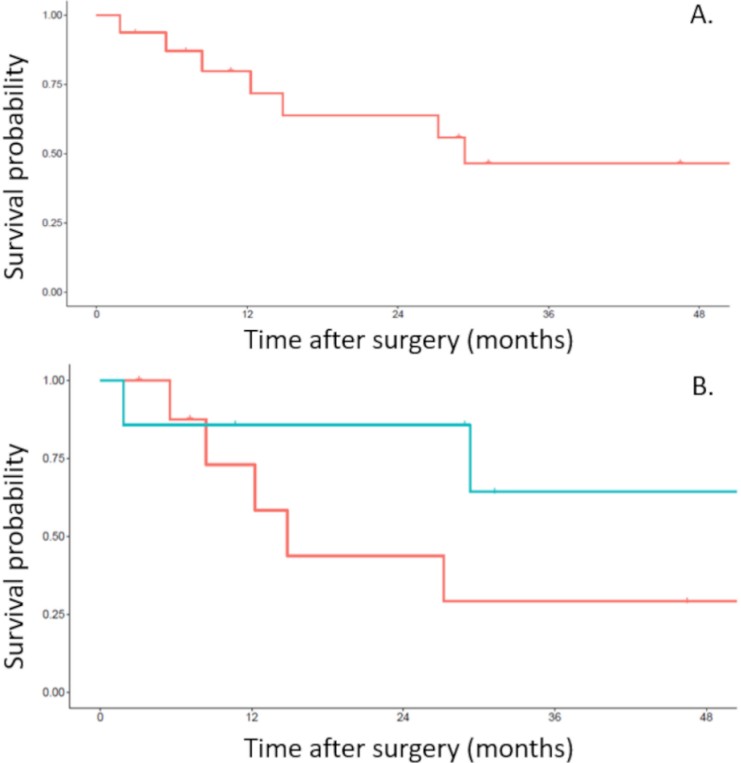

**Figure 3.** Disease free survival from the date of en bloc resection of the upper lobe, chest wall and vertebra. (**A**). Disease free survival of the whole population of the study. (**B**). Disease free survival related to the one-step (red line) or two-step (blue line) approach.

## 4. Discussion

Locally advanced non-small cell lung cancer has long been considered unresectable with poor prognosis. In the early 1950s, Chardack and MacCullum reported prolonged survival after en bloc surgical resection of a Pancoast tumor followed by adjuvant irradiation [6]. About ten years later, Shaw and Paulson demonstrated that neoadjuvant radiotherapy and extended en bloc resection with partial vertebrectomy was feasible, relatively safe and appeared to yield a long period of survival in some cases [7]. In 2001, Rusch et al. reported in a phase II trial that induction chemoradiation followed by surgical resection showed a significantly higher survival rate than did other pre-existing treatment modalities [8]. Increased experience with multimodal induction treatment facilitated resection with negative margins and led to a better long-term survival [3]. A systematic literature review yielding data on 135 patients suggested that multimodal treatment including en bloc resection provides satisfactory long-term survival [9]. In this review, overall 5-year survival rates were 43%, completeness of resection being the most significant prognostic factor. On the other hand, advances in spine surgery and collaboration with neurosurgeons allowed enlarged en-bloc vertebral resection [10]. Given these results, thoracic approaches have been refined for the resection of tumors involving the spine.

Since Shaw and Paulson in 1961, surgical techniques have evolved. The development of an anterior transcervical approach described by Cormier in 1970 [11] and subsequently refined by Dartevelle [12] in 1993 allowed the resection of locally advanced non-small cell lung cancer invading cervical structures of the thoracic inlet. The disadvantage of such a surgical approach was the need for a posterior thoracotomy if chest wall resection beyond the second rib was required. So, at this time, en bloc resection of lung cancer invading the spine combined three steps [13]: first an anterior cervical approach allowing cervical structures dissection; secondly, a classic posterolateral thoracotomy was performed for lobectomy, lymphadenectomy and ribs section with tumor-free margins. Then, the last step consisted of an enlarged posterior approach to perform vertebrectomy and en bloc resection was completed. Most recently, Chadeyras and colleagues published the evolution of their management of vertebral en bloc resection for lung cancer [14]. With increased experience and as years have gone by, the classic posterolateral thoracotomy has been abandoned. The procedure found itself simplified with "only" two approaches left: the posterior approach to the spine and the transmanubrial anterior approach. For tumors invading the spine in the middle or lower part of the chest, pulmonary resection in a lateral decubitus position and vertebrectomy in a prone position remain the standard of care. Thus, intraoperative changes in body position are usually required in patients with non-small cell lung cancer invading the spine and changes from supine or lateral position to prone are known to cause hemodynamic changes such as reduced stroke volume and cardiac index, raised central venous pressure and low blood pressure [15]. Moreover, transient loss of monitoring and possibly mobilization of the two lumen endotracheal tube causing tracheal irritation, precipitating cough, bronchospasm and pulmonary aspiration can be seen. In our single step surgical patients, hemodynamic changes were probably accentuated by inherent blood loss due to the first step lung and vertebral surgery. Consequently, to avoid body mobilization, some centers recommended a single posterior approach for advanced Pancoast tumors involving the spine [16]. As with others, our practice evolved with time. As radical en bloc resection with vertebrectomy and subsequent reconstruction is technically demanding, we progressively headed our practice towards two step surgeries performed in two separated interventions with the collaboration of spine surgeons. Moreover, this surgical strategy permitted the splitting of a long operative time that requires endurance into two surgeries that are more easily practicable. Now, for us, the posterior approach is the first surgical stage. It allows the spine surgeon to perform the osteotomy with more security with regard to neural elements, ligate the roots if needed, to control the spinal canal and to perform spinal stabilization. In the second step, the tumor was resected through a posterolateral thoracotomy or an anterior approach, which also permitted placement of anterior rods if complete vertebrectomy was required. A short delay between the two

steps permitted the completion of the en bloc resection without encountering problematic adhesive changes at the site of operation. In addition, posterior spinal arthrodesis is easier to perform with a standard posterior median approach, with the patient in prone position, than with a Paulson approach, for example. With these changes, we observed a trend towards fewer perioperative complications and a shorter mean hospital length of stay.

Results from our series suggest that vertebral en bloc resection for non-small cell lung cancer is feasible with good local tumor control and promising long term survival results at the cost of a significant morbidity. We observed a high rate of major complications that is at the upper limit of what is reported in the literature. Morbidity rates vary from 31% to 53% [14–18]. As we observed, the most frequent complications reported are pneumonia, prolonged mechanic ventilation, arrhythmia, surgical revision due to empyema or wound infection and cerebrospinal fluid leak. Local complications such as wound dehiscence and infection or CSF leakage may be favored by chemoradiation and the lengthy, hemorrhagic nature of the surgery. In our study, one patient died 56 days post operatively from pneumonia progressing to ARDS and septic shock driving our post-operative mortality rate to 6.25%. This rate confirms that published in previous series. Post-operative mortality rates published in the literature vary from 0% to 8.7% [3,14,17,18]. Post-operative deaths reported were related to pneumonia and broncho-pleural fistula. Long-term survivals observed in our small retrospective series do not allow us to make firm conclusions. Overall 3-year survival for our series was 66%. Previous series demonstrated an overall 3-year survival in the area of 53–58% [3,17]. The overall survival seemed to be higher in the two-step surgery group (40% vs. 86%). Five-year overall survival in these patients varies from 14% to 47% [4,19,20]. Although poor, overall 5-year survival in operated patients is noticeably better than non-surgical patients. Patients who received definitive chemo-radiotherapy without surgery were reported to have a 15 months overall median survival and, in that case, 5-year survival of T4N0/1 patients was 17% [21]. The disease free survival rate in our study was 55% at 3 years. The 3-year disease free survival rate was 29% in the one-step surgery group and 86% in the two-step surgery group. Previous series demonstrated a 3-year overall disease free survival at 30% [16] which is closer to what we observed in the one-step surgery group. We surprisingly observed a higher disease free survival rate in the two-step surgery group. We cannot explain this by the completeness of resection, as all patients in both groups were R0 in our study. However, this tendency toward better survival and less relapse may be favored by the growing experience of the surgical team with better patient selection since the two steps patients group was operated on more recently and through two shorter surgeries. Moreover, the two weeks period between surgeries allowed patients to recover and to perform physiotherapy explicating the better tolerance of the staged procedure. Anyway, our study has several limitations, mainly because it is a retrospective observational single-institutional study and the number of cases is too small to allow for meaningful statistical analysis.

## 5. Conclusions

Locally advanced non-small cell lung cancer invading the spine can be resected at the cost of a technically demanding surgery after induction chemoradiation treatment. Although our practice has been improved by a two-step surgery with the collaboration of spine surgeons, post-operative complications remain significant. Short-term morbidity was high but most complications were manageable. As good long-term survivals can be achieved, particularly in the two-step surgery group, this strategy of treatment should be proposed to highly selected patients, in specialized centers and after tumor board discussions.

**Author Contributions:** Conceptualization, F.T.; methodology, Y.B.; validation, F.T.; formal analysis, Y.B.; investigation: L.D.; resources, T.B.; data curation, G.D.; writing—original draft preparation, G.D.; writing—review and editing, J.M.M., R.G., G.V. and C.S.; supervision, F.T. All authors have read and agreed to the published version of the manuscript.

**Funding:** This research received no external funding.

**Institutional Review Board Statement:** The study was conducted in accordance with the Declaration of Helsinki as revised in 2013 and informed consent was received from all the patients. In accordance with French legislation, submission to an ethics committee was not required owing to the observational nature of the study.

**Informed Consent Statement:** Informed consent was received from all the patients.

**Data Availability Statement:** All data generated or analyzed during this study are included in this article. Further enquiries can be directed to the corresponding author.

**Conflicts of Interest:** G.D. reports personal fees from ASTRA ZENECA outside the submitted work. The other authors have no conflict of interest to declare. None of the authors serves as a current Editorial Team member for this journal.

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
