# Peer review of "Evolution of the Surgical Management of Lung Cancer Invading the Spine: A Single Center Experience"

_curroncol, doi:10.3390/curroncol29050248_

Round 1

Reviewer 1 Report

Thanks for the possibility to review this interesting paper.

Authors refer about their experience in the management of lung cancer invading spine and superior thoracic outlet.

We know that these patients deserve special attention for the correct choice of the surgical approach, because of particular interested anatomic district.

Often these are extremely selected patients and a thoracic surgeon is not often faced with these challenging cases.

I thank the authors for sharing their technique and their experience in the field.

I have some questions.

First of all, in the “study design” paragraph, they assert that electroneuromyography and mediastinoscopy were not routinely performed but only in selected cases. Why do you perform mediastinoscopy? Don’t you have the possibility to perform bronchoscopy and EBUS?

The second point was a curiosity: how and why did you begin to perform two-steps surgery? Because of too many peri-operative complications in one-step surgery?

After the first step, have you ever met any intra-operative trouble during second step (i.e. posterior post-operative adhesions, near-spine hematomas); have you ever had real problems in the dissection of the vertebrae wall (already dissected during first-step), during the lung surgery? If yes, how do you think to manage these intra-operative problems? Do you recommend the presence of spine surgeon during the second-step too?

I think that two-weeks period is reasonable to allow the correct recovering of the patients. Have you ever had post-operative (first-step) complications (i.e. ARDS, MOF, necessity of prolonged intubation), preventing the second operation in the next two weeks. How you should handle this situation in your opinion?

I think that this is a well-written paper. However I agree with the authors: the cohort is too small (9 vs 7) to draw conclusions but I think that their intention is to report their technique and to show that their technique is safe, feasible and even better than traditional one-step surgery.  This last aspect should be corroborated with further and larger studies, even if it is hard in the short-term (because of the rarity of these patients).

Author Response

Dear reviewer,

Please find attached our revised manuscript entitled “Evolution of the surgical management of lung cancer invading the spine: a single center experience” authored by Gabrielle Drevet, Théo Broussole, Yanis Belaroussi, Lucie Duponchelle, Jean Michel Maury, Renaud Grima, Gualter Vaz, Clément Silvestre and François Tronc. We gratefully thank the reviewer and editor for their critical and helpful comments that helped us to improve our manuscript. We here join a point-by-point response and have made the changes requested.

“Thanks for the possibility to review this interesting paper.

Authors refer about their experience in the management of lung cancer invading spine and superior thoracic outlet.

We know that these patients deserve special attention for the correct choice of the surgical approach, because of particular interested anatomic district.

Often these are extremely selected patients and a thoracic surgeon is not often faced with these challenging cases.

I thank the authors for sharing their technique and their experience in the field.

I have some questions.

First of all, in the “study design” paragraph, they assert that electroneuromyography and mediastinoscopy were not routinely performed but only in selected cases. Why do you perform mediastinoscopy? Don’t you have the possibility to perform bronchoscopy and EBUS?

Electroneuromyography and mediastinoscopy were performed in case of clinical doubt. Electroneuromyography was performed if there was any doubt about peripheral nerve involvement and mediastinoscopy was performed in case of suspicious adenopathy on PET scan. We have the possibility to perform bronchoscopy and EBUS but like mediastinoscopy, they are only performed in case of suspicious adenopathy on PET scan.

Clarifications have been made line 66-69.

The second point was a curiosity: how and why did you begin to perform two-steps surgery? Because of too many peri-operative complications in one-step surgery?

Indeed, we changed our surgical management following numerous intraoperative complications, in particular, a hemodynamic instability at the change of position, between ventral decubitus for the posterior approach and lateral decubitus for the pulmonary time. This hemodynamic instability concomitant with the change of position was probably accentuated by the consequent blood loss inherent to this type of surgery. From this observation, we came up with the idea of a 2-step surgery, leaving a few days for the patient to recover from the first operation. We then realized that the operation was better tolerated by patients and that the operation also became more comfortable for the operators.

This point has been highlighted line 48-51, line 276-277 and line 281-284.

After the first step, have you ever met any intra-operative trouble during second step (i.e. posterior post-operative adhesions, near-spine hematomas); have you ever had real problems in the dissection of the vertebrae wall (already dissected during first-step), during the lung surgery? If yes, how do you think to manage these intra-operative problems? Do you recommend the presence of spine surgeon during the second-step too?

We never encountered any per operative difficulties during the second step, such as adhesions or hematomas. We also did not observe any particular difficulty in dissection. At 15 days after the first step, very few adhesions had formed and did not interfere with the second intervention. Even though we did not encounter any particular difficulties, the neurosurgeon who performed the posterior approach is systematically present for the second step.

This information has been added line 176-177.

I think that two-weeks period is reasonable to allow the correct recovering of the patients. Have you ever had post-operative (first-step) complications (i.e. ARDS, MOF, necessity of prolonged intubation), preventing the second operation in the next two weeks. How you should handle this situation in your opinion?

We did not observe any complications such as ARDS, MOF or necessity of prolonged intubation. The main complications we observed after the first step were mainly neurological, in particular sensory deficits. This type of complication does not prevent the realization of the second step within 15 days. If such a situation were to occur, we would opt for the best compromise between a satisfactory clinical condition of the patient and an operation date not too far from the first step to avoid the problems you mentioned in your previous question.

This information has been added line 175-176.

I think that this is a well-written paper. However I agree with the authors: the cohort is too small (9 vs 7) to draw conclusions but I think that their intention is to report their technique and to show that their technique is safe, feasible and even better than traditional one-step surgery.  This last aspect should be corroborated with further and larger studies, even if it is hard in the short-term (because of the rarity of these patients).

We sincerely hope that the revised manuscript will now meet your quality standards and will be judged suitable for publication in Current Oncology.

Yours faithfully,

Gabrielle Drevet, M.D.

Department of Thoracic surgery, lung and heart-lung transplantation

Louis Pradel Hospital

Hospices Civils de Lyon

Reviewer 2 Report

Dr. Drevet and colleagues reported an interesting report of NSCLC involving the spine.

The manuscript is well written and authors should be commended for their work.

I have some points that I think they should be addressed:

  • The rationale of a two-step surgery compared to one step should be better explained.
  • What was the extent of pN2? (how many stations).
  • Was there any difference in survival between N0 and N2 patients?
  • Was the difference in survival between one and two step significant?
  • Was there any significant difference in postoperative morbidity according to the number of ribs and/or vertebrae resected?

Other minor points:

  • In the result section authors should delate the template of the manuscript (line 135-137).
  • Minor spelling typos should be reviewed.

Author Response

Dear Editor,

Please find attached our revised manuscript entitled “Evolution of the surgical management of lung cancer invading the spine: a single center experience” authored by Gabrielle Drevet, Théo Broussole, Yanis Belaroussi, Lucie Duponchelle, Jean Michel Maury, Renaud Grima, Gualter Vaz, Clément Silvestre and François Tronc. We gratefully thank the reviewer and editor for their critical and helpful comments that helped us to improve our manuscript. We here join a point-by-point response and have made the changes requested.

“Dr. Drevet and colleagues reported an interesting report of NSCLC involving the spine.

The manuscript is well written and authors should be commended for their work.

I have some points that I think they should be addressed:

The rationale of a two-step surgery compared to one step should be better explained.

Our reflection came following numerous intraoperative complications, in particular, a hemodynamic instability at the change of position, between ventral decubitus for the posterior approach and lateral decubitus for the pulmonary time. This hemodynamic instability concomitant with the change of position was probably accentuated by the consequent blood loss inherent to this type of surgery. From this observation, we came up with the idea of a 2-stage surgery, leaving a few days for the patient to recover from the first operation. We then realized that the operation was indeed better tolerated and that from a practical point of view, the operation also became more comfortable for the operators with a long surgical time split in 2.

This point has been highlighted line 48-51, line 276-277 and line 281-284.

What was the extent of pN2? (how many stations).

Two patients were pN2. These 2 patients had only one N2 station involvement.

Clarification has been made line 180.

Was there any difference in survival between N0 and N2 patients?

It is difficult to have significant results with this small number of patients, but we can nevertheless note that these 2 patients died following the procedure. One patient recurred and died within the year after surgery, the other patient died 2 years after the intervention.

This information has been added on line 219-220.

Was the difference in survival between one and two step significant?

Here again, it is difficult to obtain significant results but we can observe a trend towards better survival and less recurrence. We observe a 3-year overall survival of 40% in the one-step surgery group versus 86% in the two-step surgery group. The 3-year disease free survival rate was 29% in the one-step surgery group and the 3-year disease free survival rate was 86% in the two-step surgery group.

This has been better clarified on line 215-216.

Was there any significant difference in postoperative morbidity according to the number of ribs and/or vertebrae resected?

We did not observe any particular relationship between the number of resected ribs or vertebrae and the type or severity of complications.

This information has been added line 206-207.

Other minor points:

In the result section authors should delate the template of the manuscript (line 135-137).

Minor spelling typos should be reviewed.

Changes have been made in the text.

We sincerely hope that the revised manuscript will now meet your quality standards and will be judged suitable for publication in Current Oncology.

Yours faithfully,

Gabrielle Drevet, M.D.

Department of Thoracic surgery, lung and heart-lung transplantation

Louis Pradel Hospital

Hospices Civils de Lyon

Reviewer 3 Report

The manuscript describes a meta-analysis aimed to evaluate the efficacy and safety of surgery in lung cancer invading the spine. Indeed, the analyses conducted in this study do not add any novel information and data to the existing one. As a matter of fact, it is well know surgery in lung cancer invading the spine appears to be a safe and valid option. In addition The described data have been obtained from a  very sample size but analyzed with a good  statistical approach. However the workflow has been reported in a very synthetic manner, sometimes even too inaccurate, resulting into an inadequate respect to the aims of the article. In conclusion,  concern  is related  to the main topic of this manuscript which have been already evaluated and there are several studies all among the literature that come to the same conclusion of this study. Although the statistical analyses can be appropriated for the purpose of the study the sample size is definitely not sufficient to obtain reliable data. Therefore the illustrated results are not corroborated because of this limitation.   

Author Response

Dear Editor,

Please find attached our revised manuscript entitled “Evolution of the surgical management of lung cancer invading the spine: a single center experience” authored by Gabrielle Drevet, Théo Broussole, Yanis Belaroussi, Lucie Duponchelle, Jean Michel Maury, Renaud Grima, Gualter Vaz, Clément Silvestre and François Tronc. We gratefully thank the reviewer and editor for their critical and helpful comments that helped us to improve our manuscript. We here join a point-by-point response and have made the changes requested.

“The manuscript describes a meta-analysis aimed to evaluate the efficacy and safety of surgery in lung cancer invading the spine. Indeed, the analyses conducted in this study do not add any novel information and data to the existing one. As a matter of fact, it is well know surgery in lung cancer invading the spine appears to be a safe and valid option. In addition The described data have been obtained from a very sample size but analyzed with a good  statistical approach. However the workflow has been reported in a very synthetic manner, sometimes even too inaccurate, resulting into an inadequate respect to the aims of the article. In conclusion, concern is related  to the main topic of this manuscript which have been already evaluated and there are several studies all among the literature that come to the same conclusion of this study. Although the statistical analyses can be appropriated for the purpose of the study the sample size is definitely not sufficient to obtain reliable data. Therefore, the illustrated results are not corroborated because of this limitation. 

It is true that surgery for lung cancer invading the spine has already been described, but we wanted to focus on the change in our surgical management to a 2-step intervention, better tolerated by patients. This was confirmed by our analysis even though our study included few patients. However, it is difficult to have a large series for indications that are infrequent in thoracic surgery.

We sincerely hope that the revised manuscript will now meet your quality standards and will be judged suitable for publication in Current Oncology.

Yours faithfully,

Gabrielle Drevet, M.D.

Department of Thoracic surgery, lung and heart-lung transplantation

Louis Pradel Hospital

Hospices Civils de Lyon

Round 2

Reviewer 1 Report

I'm satisfied with the author's changes.

Reviewer 3 Report

I would like to thank the authors for addressing my questions